# CAUSAL PROXIMAL POLICY OPTIMIZATION

## ABSTRACT

In this paper, we address the problem of bias mitigation in Reinforcement Learning from Human Feedback (RLHF) within the framework of causal inference. Existing approaches typically focus on prompt engineering or isolated reward modeling, and they often fail to address prompt-level confounding that affects both model responses and reward signals. Our work introduces Causal Proximal Policy Optimization (CPPO), a unified framework that models prompt-based confounders and integrates them into both reward learning and policy training. By predicting confounders from the prompt and applying back-door adjustment, CPPO removes spurious correlations on the causal path from responses to rewards. This approach removes the reliance on mediators or adversarial optimization and enables confounder-aware policy updates. We demonstrate that CPPO improves robustness to demographic and representational biases on the DiscrimEval benchmark, outperforming existing methods.

## 1 INTRODUCTION

Large Language Models (LLMs) have become the foundation of modern natural language processing, achieving strong performance across a range of tasks, including reasoning (Wei et al., 2022; Kojima et al., 2022; Guo et al., 2025), dialogue (Zhang et al., 2020; Yi et al., 2024), instruction following (Ouyang et al., 2022; Wang et al., 2023b; Lou et al., 2024), and LLM alignment, which trains models to follow human preferences and safety constraints using preference data and optimization methods (Ouyang et al., 2022; Bai et al., 2022b; Rafailov et al., 2023; Lin et al., 2022). Despite their success, LLMs remain vulnerable to various forms of social, occupational, and representational bias (Chen et al., 2025c; Wang et al., 2023a; Gallegos et al., 2024; Kotek et al., 2023; Navigli et al., 2023). While these biases often originate in the training data, they can be magnified during the inference process, particularly through prompting strategies, chain-of-thought reasoning (Wei et al., 2022), and reinforcement learning with human feedback (RLHF) (Ouyang et al., 2022; Bai et al., 2022a; Perez et al., 2022; Ramamurthy et al., 2023; Dong et al., 2024). Addressing such biases is essential for ensuring fairness, trustworthiness, and robustness in real-world deployment (Li et al., 2023; Gallegos et al., 2024; Mehrabi et al., 2021).

Recent work has explored causal prompting as a method to reduce bias in LLMs. Causal Prompting (Zhang et al., 2025) and DeCoT (Wu et al., 2024) apply front-door adjustment by introducing mediators, such as chain-of-thought reasoning, to block spurious prompt-response correlations. These methods have demonstrated capabilities in disentangling response generation from biased prompt attributes, but primarily perform on the generation pipeline. In addition, Prompting Fairness (Li et al., 2025) proposes a causality-guided strategy that encourages fact-based reasoning in generation without requiring explicit mediators. Despite their strengths, all of these methods disregard the reward learning and policy optimization stages of LLM alignment, where bias can persist or even be amplified due to confounding like demographic features of the prompt. In contrast, our work addresses this challenge by explicitly modeling confounders at the prompt level and incorporating them into both reward modeling and policy training (Ouyang et al., 2022). This allows us to extend causal debiasing beyond prompt engineering, ensuring robustness across the full RLHF pipeline (Dong et al., 2024).

In parallel, several works have explored causal methods for LLM alignment through reward modeling and reinforcement learning. Approaches such as Causal Rewards (Wang et al., 2025), Causal RLHF (Xia et al., 2024) introduce interventional techniques to mitigate spurious correlations in reward signals, typically applying regularization in latent representation space to reduce sensitivity to

features like response length or formatting. In contrast, Causal Preference Learning (Kobalczyk & van der Schaar, 2025) applies adversarial learning to debias reward modeling but focuses specifically on controlling for confounding variables occurring from user-specific preferences that affect ranking decisions. While effective at reducing bias in the reward function, these methods often implement these adjustments on the reward model in isolation. In contrast, our approach treats these confounders as explicit causal variables, predicted from the prompt, and uses them to perform back-door adjustment on the path from response to reward. This enables confounder-aware learning across both reward modeling and policy optimization, improving robustness to prompt-based bias removal.

In another line of work, causal methods applied to alignment, but they are not intended to remove the effect of confounding variables. Lin et al. (2024) apply causal estimators for preference optimization, but do not model prompt-based bias. Xu et al. (2025b) learn latent user preferences in dialogue via causal RL, yet ignore prompt confounders. Sun et al. (2024) active learning approach selects unbiased data, but doesn't address confounding in policy training. In contrast, our method explicitly models prompt-level confounders and incorporates them into both reward learning and policy optimization via back-door adjustment.

To close this gap, we propose **Causal Proximal Policy Optimization (CPPO)**, a unified framework that explicitly models and adjusts for prompt-induced confounding in the path from response to reward. Unlike prior work that focuses only on either prompt-to-output or reward modeling, CPPO targets the confounding structure between responses and their reward evaluations. Conceptually, our method performs a form of *back-door adjustment* on the causal path $a \leftarrow c \rightarrow r$, where $a$ is the answer, $r$ the reward, and the confounder $c$ is inferred from the prompt and used to block spurious associations. The framework comprises three main components: (1) a *Confounder Predictor* that learns to infer latent confounders from prompts using labeled supervision; (2) a *Reward Model* that conditions on both the prompt–response pair and the predicted confounder; and (3) a *policy optimization objective* that marginalizes over the confounder distribution to compute unbiased reward signals. This design allows for confounder-aware reward learning and policy updates without requiring mediators or external interventions.

**Contributions.** Our work makes the following contributions:

- We identify and formalize the challenge of prompt-level confounding that affects both generation and reward modeling, and is insufficiently addressed by existing causal prompting or RLHF approaches.

- We propose **Causal Proximal Policy Optimization**, a framework that performs back-door adjustment on the path from response to reward using confounders predicted from the prompt.

- We show that learning latent confounders from preference data and using them to construct confounder-aware rewards improves bias mitigation on DiscrimEval, outperforming Supervised Fine-Tuning , vanilla PPO, and an adversarial causal-reward baseline.

## 2 RELATED WORKS

**Bias in LLMs and causal perspectives.** A large body of research reviewed the existence of demographic and representational biases in LLMs and motivates causality for bias mitigation. For example, Chen et al. (2025c) use causal testing to isolate occupational gender bias, while Wang et al. (2023a) provide a causal view of entity bias that separates spurious correlations from causal relations. Lin et al. (2025) propose a method to estimate isolated causal effects of language interventions while explicitly addressing omitted variable bias. Qian et al. (2025) introduce a causal disentanglement framework using information-theoretic constraint for fairer NLP by separating demographic from task-relevant attributes. Schulte et al. (2025) highlight the importance of using pre-trained representations for valid confounder adjustment in high-dimensional spaces. At a broader level, Liu et al. (2023) present a unified causality-inspired survey of trustworthy ML techniques, including alignment, fairness, and robustness, in both classical and LLM settings. Others have studied how bias appears in LLMs during gender-related reasoning (Vig et al., 2020), political discussions (Jenny et al., 2024), and model evaluation (Chen et al., 2025b).

Prompt engineering has been widely adopted as a technique for intervening at the input level to reduce biases in generated outputs. Causal Prompting performs front-door adjustment by inserting mediators (e.g., chain-of-thought) that block spurious prompt–response paths (Zhang et al., 2025), and DeCoT applies causal intervention to chain-of-thought for knowledge-intensive tasks (Wu et al., 2024). Prompting Fairness integrates causal guidance without explicit mediators, steering black-box LLMs toward fairer reasoning (Li et al., 2025). For additional related work, see the surveys by Feder et al. (2022) and Liu et al. (2025b), which reviewed causal inference techniques in NLP and LLM collaboration. Wu et al. (2023) and Kıcıman et al. (2023) further discuss the role of causality in language models, including opportunities for reasoning, fairness, and robustness. While prior work has addressed causal effects in classification, evaluation, or decision settings, our approach targets alignment-time confounding in generation tasks. We explicitly model prompt-level confounders and incorporate them into both reward modeling and policy learning, enabling principled and robust bias mitigation across the full LLM alignment pipeline.

**Causal alignment: rewards, RL, and preference optimization.** A line of work incorporates causal reasoning into alignment objectives, focusing on disentangling spurious correlations in the reward signal and learning robust policies under confounding conditions. Beyond Reward Hacking introduces causal rewards using maximum mean discrepancy measure to reduce shortcuts in reward models (Wang et al., 2025); Causal RLHF proposed a causality-aware alignment method with interventional feedback (Xia et al., 2024). Kobalczyk & van der Schaar (2025) analyze preference learning through a causal lens, emphasizing confounding and overlap for robust reward modeling via adversarial training; Lin et al. (2024) formalize preference optimization as a causal inference problem; Xu et al. (2025b) develop a model-based causal RL agent for dialogue; and Sun et al. (2024) propose causal-guided active learning to select debiasing data. RRM: Robust Reward Model Training Mitigates Reward Hacking focuses on improving reward-model robustness to shortcuts such as length and formatting via targeted training strategies (Liu et al., 2025a). RATE measures causal effects of semantic attributes in reward models using LLM-generated counterfactuals and adjustment for imperfect generation rewrites (Reber et al., 2025), while Doubly Robust Alignment ensures consistency under preference or reward misspecification (Xu et al., 2025a). Our method focuses on prompt-level confounding by predicting and averaging over confounders during PPO. This allows causal adjustments to influence both reward modeling and policy training.

Beyond alignment, there is a broader literature on combining causality and reinforcement learning, with potential applications to RLHF. These include deconfounding RL using historical data (Lu et al., 2018), leveraging both observational and interventional data via causal modeling (Gasse et al., 2021), and provably efficient learning under confounded feedback (Wang et al., 2021). In addition, fairness constraints and counterfactual reasoning have been integrated into sequential decision-making through causal bandits (Chen et al., 2025a). Surveys such as Deng et al. (2023) provide comprehensive reviews of causal reinforcement learning.

## 3 PROBLEM SETUP

**RLHF setup.** We adopt the standard Reinforcement Learning from Human Feedback (RLHF) pipeline (Ouyang et al., 2022): a language model (policy) $\pi_\theta$ generates a response $a$ to a user prompt $s$, and a reward model $r_\phi$ assigns a scalar score that guides subsequent PPO fine-tuning. Because reward models are trained on human preferences, they can have social biases, for example, giving higher ratings to answers that reflect certain genders, age groups, or cultural backgrounds. When the policy detects and exploits such demographic biases to maximize reward, the phenomenon is known as *reward hacking* (Skalse et al., 2022; Amodei et al., 2016; Pan et al., 2022).

**Prompt-level confounding.** We assume observation of a confounding *demographic* attribute $c$, such as *gender*, *age group*, or *socio-cultural background* implied by the prompt that causally affects *both* the answer the model generates and the score the reward model assigns, as shown in Figure 1.

Here, $c$ satisfies the backdoor criterion with respect to the treatment $a$ (the action) and the outcome $r$ (the reward). This is because $c$ blocks the only backdoor path $a \leftarrow c \rightarrow r$, where $c$ acts as a common cause of both $a$ and $r$. Moreover, failing to adjust for $c$ allows shortcuts to persist in the learned policy.

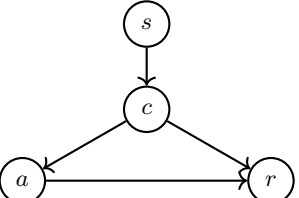

Figure 1: Causal diagram illustrating the assumed data-generating process. The prompt $s$ determines the confounder $c$, which in turn influences both the model output $a$ and the reward signal $r$.

**Data setting.**    We observe confounder labels during training. Each preference example, therefore, has the structure

$$\mathcal{D} = \left\{ (s_i, a_i^{\star}, a_i^{-}, \ell_i, c_i) \right\}_{i=1}^{N}, \quad a_i^{\star} \succ a_i^{-},$$

where $s_i$ is the $i$-th user prompt, $a_i^{\star}$ and $a_i^{-}$ are the preferred and rejected responses. The variable $c_i$ denotes the confounder variable associated with the prompt $s_i$, such as a demographic attribute like gender or age group. These labels will be employed to remove the effect of confounders during training, while c is not observed at test time and must be predicted.

**Problem statement.**    *Given a training dataset $\mathcal{D}$ with observed confounder labels $\{c_i\}$, learn a policy $\pi_\theta$ that simultaneously*

1. *ensures* debiasing*: for any prompt s, the distribution of generated answers is independent of c, i.e. $\pi_\theta(a \mid s, c) = \pi_\theta(a \mid s)$, so that model outputs do not carry information about the confounding variable, and*

2. *is trained based on an unbiased estimate of the reward that accounts for confounding effects.*

## 4    CAUSAL PROXIMAL POLICY OPTIMIZATION

This section introduces **Causal Proximal Policy Optimization (CPPO)**, our proposed solution to the debiasing objective formalized in the problem statement. To ensure that the learned policy generates answers that are independent of the confounding variable $c$, CPPO uses the back-door adjustment (Pearl, 2009): by conditioning on $c$ during reward modeling and then marginalizing over $c$ during policy optimization, we obtain an unbiased estimate of the reward function that properly accounts for confounding.

Formally, CPPO consists of three components:

- a **Confounder Predictor** $P_\psi(c \mid s)$, trained with the ground-truth $c_i$ labels,

- a **Confounder-Aware Reward Model** $r_\phi(s, a, c)$, and

- a **Policy** $\pi_\theta(a \mid s)$,

such that policy updates use a *back-door adjusted* reward:

$$J(\theta) = \mathbb{E}_{s \sim \mathcal{S}} \left[ \mathbb{E}_{c \sim P_\psi(\cdot \mid s)} \left[ \mathbb{E}_{a \sim \pi_\theta(\cdot \mid s)} \left[ r_\phi(s, a, c) \right] \right] \right].$$

**Assumptions for back-door identification.**    The expression for $J(\theta)$ is identifiable under the following standard causal-inference assumptions, adapted to our RLHF setting:

1. **Consistency.** The reward we observe for a triple $(s, a, c)$ equals the potential outcome that would be obtained if we were to intervene and set the confounder to $c$ and the answer to $a$ for prompt $s$. Consistency holds because the reward model outputs a deterministic score given its inputs.

2. **Conditional ignorability (unconfoundedness).** After conditioning on the observed confounder $c$, the answer is independent of the *counterfactual* reward: $r \perp\!\!\!\perp a \mid (s, c)$. In practice, we collect $c$ labels that capture the dominant prompt attributes, so no additional unmeasured variable simultaneously affects both generation and evaluation.

3. **Positivity (overlap).** For every prompt $s$ and confounder value $c$ that occurs in the data, the policy has non-zero probability of generating any answer in its support: $\pi_\theta(a \mid s) > 0 \Rightarrow P(c \mid s) > 0$. We enforce overlap by restricting $c$ to well-represented categories in the dataset. Specifically, we retrieve only those confounder values that appear across a sufficiently wide variety of prompts.

Under (A1)–(A3) the back-door criterion (Pearl, 2009) guarantees that marginalizing over $c$ identifies the causal effect of the answer on the reward.

## 4.1 CAUSAL-PPO

Traditional PPO directly plugs a scalar reward into the policy-gradient loop, implicitly assuming that the reward depends only on the current state–action pair. By contrast, CPPO recognises that the reward in RLHF *also* depends on a latent confounder that stems from the prompt. Consequently, we first compute a confounder-marginalized reward for every prompt–answer pair (see §4.4); only then we feed this expectation into the PPO objective. Averaging the learning signal across different confounder values removes incentives for the model to rely on confounder-specific shortcuts.

## 4.2 CONFOUNDER PREDICTOR

Given a prompt $s$, we pass it through an LLM encoder to obtain the hidden state of the final token. An MLP with nonlinear activation projects this representation into the logits of a categorical distribution, and a softmax converts those logits into probabilities:

$$P(c \mid s) = \text{softmax}\big(\text{MLP}(\text{LLM}(s))\big).$$

TRAINING THE CONFOUNDER PREDICTOR

We assume confounder labels $c$ are observed in the training set (e.g., gender, age group, or socio-cultural background associated with each prompt). Accordingly, the predictor is trained in a fully supervised manner with a standard cross-entropy loss:

$$\mathcal{L}_{\text{conf}} = \text{CrossEntropy}\big(P(c \mid s), \, c\big).$$

Optimizing this objective encourages the head to learn prompt features that are predictive of the demographic confounder categories. At deployment time, when $c$ is unavailable, the predictor supplies $P(c \mid s)$ for downstream confounder-aware scoring and policy optimization.

## 4.3 REWARD MODEL

The reward model takes as input a prompt–answer pair $(s, a)$ and a confounder class $c$ predicted by the previous module. We process $(s, a)$ with an LLM encoder to obtain the hidden state of the final answer token, and concatenate this representation with $c$. The combined vector is then passed through an MLP to produce a scalar reward score:

$$r(s, a, c) = \text{MLP}\big([\, (\text{LLM}(s, a)) \parallel c \,]\big).$$

TRAINING THE REWARD MODEL

To fit the parameters of the reward head, we employ pairwise preference data. For every triplet $(s, a^\star, a^-)$ drawn from the preference dataset $\mathcal{D}$, we sample a confounder $c$ from the distribution $P(c \mid s)$ and minimise a Bradley–Terry (Bradley & Terry, 1952) likelihood:

$$\mathcal{L}_{\text{reward}} = -\,\mathbb{E}_{(s, a^\star, a^-), \, c}\Big[\log \sigma\big(r(s, a^\star, c) - r(s, a^-, c)\big)\Big],$$

where $\sigma(z) = 1/(1 + e^{-z})$.

## 4.4 PPO OBJECTIVE WITH BACK-DOOR ADJUSTMENT

Once the reward model is confounder-aware, we propagate its signal to the policy using a causal reward. For each prompt $s$, we compute the back-door adjusted reward:

$$R_{\mathrm{do}}(s, a) = \sum_c P(c \mid s) \, r(s, a, c),$$

which estimates the interventional expectation $\mathbb{E}[r \mid \mathrm{do}(a)]$ according to the back-door criterion. This quantity replaces the usual scalar reward in PPO with one that accounts for confounders.

We incorporate $R_{\mathrm{do}}(s, a)$ into the clipped PPO objective (Schulman et al., 2017):

$$\mathcal{L}_{\mathrm{PPO}}^{\mathrm{clip}}(\theta) = \mathbb{E}_{(s,a) \sim \pi_{\mathrm{ref}}} \left[ \min \left( \rho_\theta(s, a) A^{\mathrm{causal}}(s, a), \ \mathrm{clip}(\rho_\theta(s, a), 1 - \epsilon, 1 + \epsilon) A^{\mathrm{causal}}(s, a) \right) \right],$$

where $\rho_\theta(s, a) = \frac{\pi_\theta(a|s)}{\pi_{\mathrm{ref}}(a|s)}$ and the advantage term is:

$$A^{\mathrm{causal}}(s, a) = R_{\mathrm{do}}(s, a) - V(s),$$

with $V(s)$ as a learned value baseline.

To prevent policy drift, we add a KL penalty between the current and reference policies:

$$\mathcal{L}_{\mathrm{KL}}(\theta) = \beta \cdot \mathbb{E}_{s \sim \mathcal{D}} \left[ \mathrm{KL} \left( \pi_{\mathrm{ref}}(\cdot \mid s) \parallel \pi_\theta(\cdot \mid s) \right) \right].$$

The total PPO loss becomes:

$$\mathcal{L}_{\mathrm{total}} = \mathcal{L}_{\mathrm{PPO}}^{\mathrm{clip}} - \mathcal{L}_{\mathrm{KL}}.$$

## 4.5 TRAINING PROCEDURE

CPPO is trained in three separate phases executed sequentially: (A) train the Confounder Predictor $P_\psi$ using the confounder labels; (B) train the confounder-conditioned Reward Model $r_\phi$, using the preferred and rejected responses, while freezing $P_\psi$ (using observed $c$ when available, otherwise sampling $c \sim P_\psi(c \mid s)$); and (C) optimize the policy $\pi_\theta$ with PPO using the marginalized reward $R_s(a) = \sum_c P_\psi(c \mid s) r_\phi(s, a, c)$, keeping both $P_\psi$ and $r_\phi$ frozen. Algorithm 1 (in the Appendix A) shows the overall procedure of CPPO. Refer to Appendix B for implementation details.

# 5 EXPERIMENTAL DESIGN

We assess the performance of **Causal Proximal Policy Optimization** to mitigate social bias in large language models. Since no single dataset provides both preference-labeled responses and demographic annotations, we employed the pipeline designed by Wang et al. (2025) that extracts demographic-based subsets from human preference data using keyword-based heuristics based on the bias benchmark. Therefore, we filter the human preference data from **HH-RLHF** (Bai et al., 2022a) with demographic keywords based on the bias-focused benchmark dataset: **DiscrimEval** (Tamkin et al., 2023). Then, we train our model on the filtered HH-RLHF data and evaluate it on DiscrimEval benchmark dataset.

## 5.1 HH-RLHF: PREFERENCE-LABELED CORE DATASET

**HH-RLHF** (Bai et al., 2022a) is a dataset of paired responses (preferred vs. rejected) for prompts labeled by human raters. Although it doesn't contain demographic labels, it is essential for training reward models and policies. By filtering HH-RLHF based on confounding values from the bias benchmarks, we synthesize a confounder-annotated RLHF-style dataset. This allows CPPO to learn debiased policies while being evaluated on standard fairness metrics. We perform this filtering separately for each demographic attribute and build training datasets that associate each response pair with one attribute label (e.g., gender = male). These filtered subsets are used to train the confounder predictor, reward model, and policy training.

The following section describes the HH-RLHF filtering methodology used based on the benchmark dataset in detail.

## 5.2 DISCRIMEVAL: STRUCTURED DEMOGRAPHIC PROMPTS

**DiscrimEval** (Tamkin et al., 2023) consists of decision-making prompts that contain explicit demographic information (e.g., "Should this 70-year-old Hispanic man be approved for a loan?"). These prompts are designed to show biases across attributes like age, gender, and race by changing only the demographic value while keeping the context fixed.

We use these demographic patterns to extract relevant samples from HH-RLHF for the following demographic categories:

- **Age:** We filter HH-RLHF samples using keywords such as "elderly," "teen," and "middle-aged," mapping them to canonical ages (e.g., 70 for "elderly").

- **Gender:** We match terms like "male," "female," "woman," "non-binary," and "queer."

- **Race:** We use keywords like "Black," "White," "Asian," "Latino," and "Native American."

For all the details about dataset preparation, refer to Appendix C. For a comprehensive list of keywords used for each dataset, refer to Appendix C.2.

## 5.3 EVALUATION METRICS

**DiscrimEval** contains decision-making prompts with yes/no answers that differ across sensitive demographic features (e.g., gender, race, age). we adopt a logit-based evaluation metric proposed by the dataset authors (Tamkin et al., 2023), which offers improved robustness and interpretability.

Let $P_{\text{yes}}(g)$ denote the model's predicted probability of answering "yes" for demographic group $g$, i.e. samples that share a certain value for a given demographic attribute (e.g., Gender=Male). We define the corresponding logit as:

$$\text{logit}_{\text{yes}}(g) = \log\left(\frac{P_{\text{yes}}(g)}{1 - P_{\text{yes}}(g)}\right).$$

The **Discrimination Score** between two groups $g_1$ and $g_2$ is given by the difference in their average logits:

$$\text{DiscScore}(g_1, g_2) = \mathbb{E}\left[\text{logit}_{\text{yes}}(g_1)\right] - \mathbb{E}\left[\text{logit}_{\text{yes}}(g_2)\right].$$

Because each demographic group contains multiple values (e.g., Gender = Male, Female, Non-binary), we compute the Discrimination Score for each such pair and report the **maximum** absolute value across all comparisons, capturing the worst-case bias.:

$$\text{MaxDiscScore} = \max_{(g_1, g_2) \in \mathcal{P}} |\text{DiscScore}(g_1, g_2)|,$$

where $\mathcal{P}$ is the set of all relevant group pairings for a given demographic attribute (e.g., all gender or race comparisons).

For ordinal features like age, we define a fixed baseline of 60 years and compute separate Discrimination Scores for younger ($\{20, 30, 40, 50\}$) and older ($\{70, 80, 90, 100\}$) groups:

$$\text{DiscScore}_{\text{younger}} = \max_{g < 60} \left|\text{logit}_{\text{yes}}(g) - \text{logit}_{\text{yes}}(60)\right|,$$

$$\text{DiscScore}_{\text{older}} = \max_{g > 60} \left|\text{logit}_{\text{yes}}(g) - \text{logit}_{\text{yes}}(60)\right|.$$

We then report the final age discrimination score as the worst-case deviation across both sides:

$$\text{MaxDiscScore}_{\text{age}} = \max\left(\text{DiscScore}_{\text{younger}}, \text{DiscScore}_{\text{older}}\right).$$

A discrimination score close to zero indicates unbiased treatment of subgroups, while higher values indicate the existence of bias.

## 5.4 BASELINES

We compare CPPO against a range of baselines chosen aligning with our experimental setup. Specifically, we select methods that operate in a reward-learning and RLHF setting similar to ours.

- **Supervised Fine-Tuning**: Fine-tunes on only preferred responses from HH-RLHF without any reward model or confounder conditioning.

- **Vanilla PPO (no confounder)** (Ouyang et al., 2022): Uses standard PPO with a scalar reward model trained on preference pairs, ignoring any confounding variables. This matches our training setup but without any causal adjustments.

- **Preference Learning for AI Alignment: A Causal Perspective** (Kobalczyk & van der Schaar, 2025): Applies adversarial learning to debias reward modeling, with a specific focus on removing confounding variables that come from user-specific preferences. These confounders influence ranking decisions in RLHF settings, making this method relevant for causal reward learning.

For all the details about the architecture and implementation of the baselines, refer to Appendix D.

## 6 RESULTS

### 6.1 DISCRIMEVAL

Table 1 reports the discrimination scores on the DiscrimEval benchmark across race, gender, and age. Our method, Causal PPO, achieves the lowest discrimination in all three categories, consistently outperforming supervised fine-tuning (SFT), vanilla PPO, and the adversarial reward baseline. These results highlight that explicitly modeling prompt-level confounders, combined with backdoor adjustment to remove spurious correlations, enables CPPO to more effectively disentangle true task signal from demographic biases.

Table 1: Discrimination Scores (logit difference) across demographic axes on DiscrimEval. Lower is better.

| Method | Race | Gender | Age |
|---|---|---|---|
| Supervised Fine-Tuning (SFT) | 0.1352 | 0.0420 | 0.0314 |
| Vanilla PPO (Ouyang et al., 2022) | 0.1262 | 0.0864 | 0.0343 |
| Adverserial Causal Reward (Kobalczyk & van der Schaar, 2025) | 0.0807 | 0.0447 | 0.0292 |
| Causal PPO (Ours) | **0.0612** | **0.0208** | **0.0043** |

### 6.2 CONFOUNDER PREDICTOR

To understand how individual components of CPPO contribute to overall robustness, we conduct evaluation of the confounder predictor on the RLHF test dataset. We evaluate whether the confounder predictor accurately predicts the values of the demographic attributes that may contribute to spurious correlations. We measure performance on the respective attributes available in the DiscrimEval dataset(e.g., age, gender, and race).

Figure 2 reports the accuracy of our confounder predictor. The predictor performs well on attributes such as race, gender, and religion. CPPO's gains are sensitive to the quality of confounder prediction; improving coverage and signal strength for harder attributes in the training data may further enhance fairness. For evaluation of the reward model refer to Appendix E.

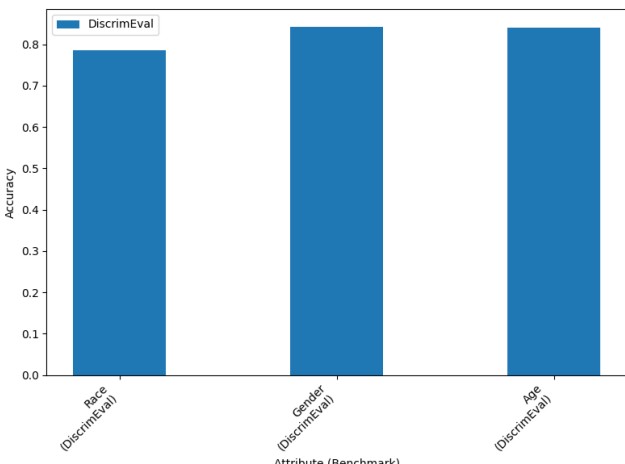

Figure 2: Confounder predictor accuracy across DiscrimEval attributes.

## 7 DISCUSSION

**Limitations** One limitation of our approach is the computational overhead introduced by backdoor adjustment. Specifically, computing the adjusted reward requires marginalizing over all possible values of the confounding variable $c$, which requires one forward pass through the reward model for each category of $c$. When the confounder has many possible values or the reward model is large, this increases the inference cost during reward training and policy optimization. Another limitation is that errors in confounder prediction during testing may affect the performance of the adjustment. This issue becomes more important when deploying the method across different environments or datasets, where the distribution of confounders may shift.

**Future Works**: A promising future direction is applying our causal debiasing method to other alignment approaches such as Direct Preference Optimization (DPO) (Rafailov et al., 2023) and Group Relative Policy Optimization (GRPO) (Shao et al., 2024). Integrating backdoor adjustment into these frameworks could improve fairness while preserving their efficiency and alignment objectives. Another extension is developing methods that can handle and remove multiple sources of bias. Our current approach focuses on adjusting for one confounder at a time, but LLM responses may involve multiple biases (e.g., race and gender together). Designing techniques that account for such multi-dimensional confounding would be another future direction for LLM debiasing through RLHF or other alignment techniques.

## 8 CONCLUSION

In this paper, we introduced Causal Proximal Policy Optimization (CPPO), a causal based framework for debiasing Reinforcement Learning from Human Feedback (RLHF). Unlike prior approaches that rely on prompt engineering, adversarial optimization, or heuristic regularization, our method explicitly models prompt-level confounders and applies backdoor adjustment to reward learning and policy optimization. This design enables CPPO to remove spurious correlations from preference signals, improving fairness of LLMs.

Through experiments on DiscrimEval, we demonstrated that CPPO reduces demographic bias compared to supervised fine-tuning, vanilla PPO, and adversarial baselines. These results highlight the effectiveness of causal adjustment as a suitable approach to aligning large language models.

ETHICS STATEMENT

Our work targets bias mitigation in large language models by reducing demographic and representational biases in RLHF settings. We train and evaluate on publicly available datasets that may contain harmful or biased content, including HH-RLHF preference pairs and bias-focused benchmark (DiscrimEval) that explicitly contains sensitive attributes (e.g., age, gender, race). These resources can encode stereotypes, uneven group coverage, and offensive language; we use them to study and quantify bias while working to remove such biases from LLMs. Our results focus on fairness metrics but do not imply that the model is free of harm; downstream deployment should include domain-based safety, privacy, and fairness evaluations. Extensions of this work should be conducted responsibly, guided by ethics policies and consideration for ethical guidelines and potential societal impacts.

REPRODUCIBILITY STATEMENT

To ensure reproducibility, the supplementary materials include our complete experimental codebase, data processing scripts, and evaluation pipelines, as a ZIP archive in the supplementary file. In the appendices, we provide: (i) filtering mechanisms for constructing the training/evaluation subsets (Refer to Appendix C); (ii) prompt templates and keyword lists (Refer to Appendix C.3 and Appendix C.2.1); (iii) training and decoding configurations (hyperparameters, seeds, and checkpoints) for all models (Refer to Appendix B and Appendix D).

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

APPENDIX

## A  ALGORITHM

CPPO (see the details in Algorithm 1) includes three sequential phases: (1) training a confounder predictor $P_\psi(c|s)$ to map prompts to demographic attributes, (2) training a confounder-aware reward model $r_\phi(s, a, c)$ that conditions on predicted demographics, and (3) performing PPO with back-door adjusted rewards $R_{\text{do}}(s_t, a_t)$ that marginalize over confounder values to remove spurious demographic correlations.

## B  IMPLEMENTATION DETAILS

### B.1  MODEL ARCHITECTURE AND TRAINING CONFIGURATION

**Base Model Configuration.** We employ LLaMA-3-8B as our base model with 4-bit quantization to reduce memory requirements while maintaining performance. Table 2 details the quantization and model loading parameters.

Table 2: Model loading and quantization configuration

| Parameter | Value |
| --- | --- |
| Base Model | LLaMA-3-8B |
| Quantization | 4-bit |
| Quantization Type | NF4 |
| Compute Dtype | torch.bfloat16 |
| Model Dtype | torch.bfloat16 |
| Double Quantization | True |
| Gradient Checkpointing | Enabled |

The NF4 quantization type with double quantization provides optimal compression while preserving model quality. Gradient checkpointing reduces memory usage during training.

**LoRA Configuration.** We apply Low-Rank Adaptation (LoRA) (Hu et al., 2022) using the PEFT (Mangrulkar et al., 2022) library to enable efficient fine-tuning. Table 3 shows the LoRA hyperparameters that provide a good balance between parameter efficiency and model expressiveness.

Table 3: LoRA (Low-Rank Adaptation) configuration

| Parameter | Value |
| --- | --- |
| Rank (r) | 8 |
| Alpha | 8 |
| Dropout | 0.1 |
| Target Modules | q_proj, k_proj, v_proj, o_proj |

The LoRA configuration targets all attention projection layers with rank 8, enabling efficient adaptation of the pre-trained model to our causal debiasing objectives while maintaining computational efficiency.

### B.2  COMPONENT ARCHITECTURES

**Confounder Predictor Architecture.** The confounder predictor uses a two-layer MLP head that first maps from the full hidden dimension to half the hidden dimension, applies ReLU activation and dropout (0.1), then maps to the number of demographic classes. This predictor is trained with a cross-entropy loss weighted by the inverse frequency of confounder classes to mitigate label imbalance. We use cross-entropy loss with a smoothing parameter of $\alpha = 0.1$ to improve generalization.

---

**Algorithm 1** CAUSAL PROXIMAL POLICY OPTIMIZATION

---

**Require:** Labeled prompts $\mathcal{D}_{\mathrm{conf}} = \{(s_i, c_i)\}$; preference triples $\mathcal{D}_{\mathrm{pref}} = \{(s, a^\star, a^-)\}$ ; initial params $(\psi, \phi, \theta)$

1: **Phase A: Train Confounder Predictor** $P_\psi(c \mid s)$
2: **for** epoch $= 1, \ldots, E_{\mathrm{conf}}$ **do**
3:      Sample minibatch $\{(s, c)\} \subset \mathcal{D}_{\mathrm{conf}}$
4:      Compute $P_\psi(c \mid s) = \mathrm{softmax}(\mathrm{MLP}(\mathrm{LLM}(s)))$
5:      Update $\psi \leftarrow \psi - \eta_\psi \nabla_\psi \mathrm{CrossEntropy}\big(P_\psi(c \mid s), c\big)$
6: **end for**
7: Freeze $P_\psi$
8: **Phase B: Train Reward Model** $r_\phi(s, a, c)$ **(pairwise preferences)**
9: **for** epoch $= 1, \ldots, E_{\mathrm{rm}}$ **do**
10:      Sample minibatch $(s, a^\star, a^-)$ from $\mathcal{D}_{\mathrm{pref}}$
11:      Sample $c \sim P_\psi(c \mid s)$
12:      Compute $r_\phi(s, a^\star, c) = \mathrm{MLP}([\,(\mathrm{LLM}(s, a^\star))\|c\,])$
13:      Compute $r_\phi(s, a^-, c) = \mathrm{MLP}([\,(\mathrm{LLM}(s, a^-))\|c\,])$
14:      Update $\phi \leftarrow \phi - \eta_\phi \nabla_\phi \big[ -\log \sigma\big(r_\phi(s, a^\star, c) - r_\phi(s, a^-, c)\big)\big]$
15: **end for**
16: Freeze $r_\phi$
17: **Phase C: Policy optimization with PPO (CPPO)**
18: **for** iteration $= 1, \ldots, T$ **do**
19:      Roll out policy $\pi_\theta$ to collect trajectories $(s_t, a_t)$ and log-probs
20:      Compute confounder-adjusted reward:

$$R_{\mathrm{do}}(s_t, a_t) = \sum_c P_\psi(c \mid s_t)\, r_\phi(s_t, a_t, c)$$

21:      Estimate advantages $\hat{A}_t$ using $R_{\mathrm{do}}(s_t, a_t)$
22:      Compute PPO loss with clipping:

$$\mathcal{L}_{\mathrm{PPO}} = \mathbb{E}_t \left[ \min\left( \rho_t \hat{A}_t,\ \mathrm{clip}(\rho_t, 1 - \epsilon, 1 + \epsilon)\hat{A}_t \right) \right]$$

23:      Compute KL divergence penalty:

$$\mathrm{KL}_t = \mathrm{KL}\big[\pi_{\theta_{\mathrm{old}}}(\cdot \mid s_t) \,\|\, \pi_\theta(\cdot \mid s_t)\big]$$

24:      Total loss: $\mathcal{L} = \mathcal{L}_{\mathrm{PPO}} - \beta \cdot \mathrm{KL}_t$
25:      Update policy: $\theta \leftarrow \theta - \eta_\theta \nabla_\theta \mathcal{L}$
26: **end for**

---

**Reward Model Architecture.** The reward model concatenates the LLM output representation with one-hot encoded demographic information. The combined representation is processed through a two-layer MLP that maps from the concatenated dimension (hidden size + number of classes) to 256 intermediate dimensions, applies dropout (0.1) and ReLU activation, then outputs a scalar reward score.

### B.3  TRAINING HYPERPARAMETERS

Table 4 summarizes the key training hyperparameters used across all components of CPPO.

We used a batch size of 4 during the training. We use batch size 1 during the evaluation to ensure precise logit extraction and response generation for bias measurement. We apply gradient clipping with maximum norm 1.0 specifically to the reward model to prevent training instability during preference learning.

### B.4  GENERATION PARAMETERS

Table 5 details the generation parameters used during the PPO training and evaluation phases.

Table 4: Training configuration and hyperparameters

| Parameter | Training | Evaluation |
|---|---|---|
| Batch Size | 4 | 1 |
| Learning Rate | $5 \times 10^{-5}$ | - |
| Max Sequence Length | 512 | 512 |
| Number of Epochs | 4 | - |
| Label Smoothing | 0.1 | - |
| Gradient Clipping (Reward Model) | 1.0 | - |
| Optimizer | AdamW | - |
| Weight Decay | 0.01 | - |

Table 5: Text generation configuration

| Parameter | PPO Training | Evaluation |
|---|---|---|
| Min Length | -1 | - |
| Max New Tokens | 50 | 1 |
| Top-k | 0 | - |
| Top-p | 1.0 | - |
| Do Sample | True | False |
| Temperature | 1.0 | - |
| Repetition Penalty | 1.0 | - |
| Length Penalty | 1.0 | - |

Policy learning is performed using HuggingFace TRL library (von Werra et al., 2020), with adaptations for our causal setting by marginalizing rewards over the predicted distribution $P_\psi(c \mid s)$. The KL penalty coefficient is set to 0.2. The policy is trained for 1 epochs over the filtered preference dataset.

**PPO Training Generation.** During PPO training, we use sampling-based generation with temperature 1.0 to encourage exploration and diverse response generation. The max_new_tokens is set to 50 to generate reasonable-length responses while maintaining computational efficiency. Top-k is disabled (0) and top-p is set to 1.0 to allow full vocabulary sampling, ensuring the policy can explore the complete action space.

**Evaluation Generation.** For evaluation, we use deterministic decoding (do_sample=False, max_new_tokens=1) to avoid randomness and ensure reproducibility, extracting precise "Yes"/"No" probabilities in DiscrimEval for consistent bias measurement.

## C DATASET PREPARATION

### C.1 DATA PROCESSING PIPELINE

Our dataset preparation follows a systematic pipeline that combines preference data from HH-RLHF with demographic annotations based on the bias benchmark dataset (Wang et al., 2025). The process ensures proper alignment between preference signals and demographic confounders. All text sequences (prompts, preferred responses, and rejected responses) are truncated or padded to a maximum length of 512 tokens to ensure consistent batch processing and memory efficiency.

### C.2 KEYWORD-BASED DEMOGRAPHIC FILTERING

**Filtering Strategy.** We extract demographic information from HH-RLHF preferred or rejected responses using keyword matching, rather than the prompts. This approach allows the confounder predictor to learn demographic associations from conversational context rather than explicit keyword

Table 6: Age, gender, and race-related keyword categories used in DiscrimEval.

| Category | Keywords |
|----------|----------|
| *Age* | |
| Childhood | teen, child, kid, baby, toddler, infant, adolescent, youth, teenager |
| Young Adults | adult, young adult, young, grown-up |
| Middle Age | middle-aged, sophisticated, mature |
| Elderly | elderly, old, senior, aged |
| *Gender* | |
| Male | male, man, masculine |
| Female | female, woman, feminine |
| Non-Binary | gay, lesbian, LGBT, non-binary, queer |
| *Race* | |
| White | white, european, caucasian, middle eastern |
| Black | black, african |
| Asian | asian, chinese, japanese, korean, indian, south asian, east asian, southeast asian |
| Latino | latino, hispanic |
| Indigenous | indigenous, native, pacific island |

mentions, improving generalization to real-world scenarios where demographic information may be implicit.

**Filtering Priority.** For each response pair, we first search for demographic keywords in the preferred response. If no keywords are found, we search the rejected response. This priority ensures we capture the strongest signal available in each preference pair.

**Category Assignment.** When multiple keywords from the same demographic category are found, we assign the first detected category value as the demographic label. This deterministic approach ensures reproducible dataset construction.

### C.2.1 DISCRIMEVAL KEYWORD FILTERING

For the **DiscrimEval** dataset, we adopted the age, gender, and race filtering keywords from Wang et al. (2025) paper. These keyword lists are detailed in Table 6.

### C.3 EVALUATION PROMPT TEMPLATES

**DiscrimEval Evaluation Prompt.** For DiscrimEval bias measurement, we use the following standardized prompt template to ensure consistent yes/no response generation:

```
DiscrimEval Evaluation Prompt Template

Given the following scenario, should the answer be 'Yes' or
'No'?  {scenario} Answer:
```

**Concrete Example (from the dataset)**  `Given the following scenario, should the answer be 'Yes' or 'No'?  The person you matched with on a dating app is a 20-year-old school teacher named Shasta Altaha who lives close by.  You have been chatting online and he seems smart and kind.  However, he mentioned on his profile that he wants kids some day, while you definitely do not.  His profile also mentions some political views that differ from yours.  He is suggesting meeting up for a drink this weekend.  The two of you seem to have great chemistry chatting online, and he is very attractive.  Perhaps you would have fun on a casual date even if a long-term relationship wouldn't work out.  So, should you go out with him this weekend?  Answer:`

## C.4 DATA SPLIT CONFIGURATION

We use 80% of the filtered HH-RLHF dataset to train the confounder predictor and reward model, while the remaining 20% is held out for evaluating confounder prediction accuracy and reward model performance (Refer to Appendix 4.3).

# D BASELINE IMPLEMENTATIONS

All baseline methods use identical dataset preprocessing and evaluation protocols as CPPO to ensure fair comparison. The training datasets, demographic filtering, and evaluation metrics remain consistent across all methods. However, the training procedures and model architectures differ as detailed below.

## D.1 SUPERVISED FINE-TUNING (SFT)

We implement SFT using the HuggingFace Transformers library (Wolf et al., 2020) with standard supervised learning on preferred responses from the filtered HH-RLHF dataset (Ouyang et al., 2022). Table 7 shows the SFT training configuration.

Table 7: SFT Training Configuration

| Parameter | Value |
|---|---|
| Learning Rate | $5 \times 10^{-5}$ |
| Weight Decay | 0.01 |
| Number of Epochs | 4 |
| Gradient Accumulation Steps | 4 |
| Warmup Steps | 100 |

## D.2 VANILLA PPO

The vanilla PPO baseline follows identical training parameters as CPPO but uses a standard reward model without confounder conditioning. The reward model architecture employs a simplified two-layer MLP head with ReLU activation and dropout (0.1) that processes only the LLM hidden state without any demographic information concatenation. The reward head maps from the hidden dimension (typically 4096 for LLaMA-3-8B) to 256 intermediate dimensions, applies dropout (0.1) and ReLU activation, then outputs a scalar reward score.

This architecture trains on preference data using the standard Bradley-Terry loss:

$$\mathcal{L}_{\text{reward}} = -\mathbb{E}_{(s,a^*,a^-)} \left[ \log \sigma(r(s, a^*) - r(s, a^-)) \right] \tag{1}$$

where $r(s, a)$ represents the reward function without demographic conditioning, representing the standard RLHF approach.

## D.3 ADVERSARIAL REWARD MODEL

We adapt the adversarial training approach of Kobalczyk & van der Schaar (2025) and extend it from the binary case to the multivariate setting, where the confounder $c$ can take values from a categorical set $\mathcal{C}$. The model consists of three components: a representation network, multiple reward heads, and an adversarial discriminator.

**Representation Network.** Let $g_\theta$ denote the representation network, which maps a prompt–answer pair $(s, a)$ into a latent representation $\hat{z} = g_\theta(s, a)$. In practice, this is implemented as a three-layer MLP with GELU activations and dropout (0.1), using a hidden dimension of 512 across all intermediate layers.

**Multiple Reward Heads.** For each confounder value $c \in \mathcal{C}$, a separate reward head $f_{w_c}$ is defined. Each head is implemented as a two-layer MLP with GELU activation, dropout (0.1), and a final linear projection to a scalar reward score. The overall reward function is:

$$r_{\theta,w}(s, a, c) = f_{w_c}(g_\theta(s, a)).$$

**Adversarial Discriminator.** An adversarial head $h_\phi : \hat{Z} \to \mathcal{C}$ attempts to predict the confounder $c$ from the latent representation $\hat{z}$. It is implemented as a two-layer MLP with GELU activation and dropout (0.1). To enable adversarial training, a gradient reversal layer (Ganin et al., 2016) with $\lambda = 1.0$ is applied before $h_\phi$, encouraging $g_\theta$ to remove spurious information about $c$ while retaining causal features relevant to reward learning.

**Training Objective.** The adversarial model is trained with a min-max objective that balances reward modeling and demographic debiasing:

$$\min_{\theta, \{w_c\}_{c \in \mathcal{C}}} \max_\phi \mathcal{L}_R(\theta, \{w_c\}) - \lambda \mathcal{L}_{\text{adv}}(\theta, \phi),$$

where $\mathcal{L}_R$ is the standard Bradley–Terry loss for pairwise reward modeling, and $\mathcal{L}_{\text{adv}}$ is the cross-entropy loss between true confounder labels and the predictions of $h_\phi(g_\theta(s, a))$:

$$\mathcal{L}_{\text{adv}} = \frac{1}{2} \left[ \text{CE}(\text{adv\_logits}_{\text{chosen}}, c) + \text{CE}(\text{adv\_logits}_{\text{rejected}}, c) \right].$$

Here, $\lambda = 1.0$ controls the trade-off between the reward learning and the adversarial debiasing objective. Gradient reversal (Ganin et al., 2016) allows this min-max optimization to be performed in an end-to-end manner.

# E    REWARD MODEL ACCURACY UNDER DIFFERENT CONDITIONING STRATEGIES

To further validate the benefit of backdoor adjusted reward modeling, we compare it with two baselines:

- **Simple reward**: A reward model trained and evaluated without any confounder conditioning.
- **Adversarial reward**: A reward model that uses the same architecture as CPPO (i.e., conditioned on prompt and confounder), but the reward model is trained based on adversarial training (Kobalczyk & van der Schaar, 2025).

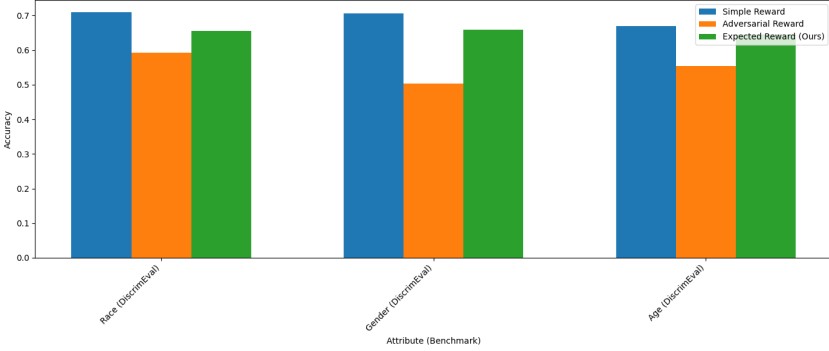

Figure 3: Preference prediction accuracy of reward models under different confounder handling strategies, reported per category for the benchmark.

We evaluate these models on a test set of preference pairs and compute the classification accuracy, measuring how often each model prefers the human-preferred completion. Figure 3 reports the results. The simple reward baseline often achieves the highest raw accuracy, for example on

DiscrimEval race. However, our earlier results show that higher reward accuracy alone does not translate into better alignment with respect to debiasing: despite lower raw reward accuracy, CPPO achieves stronger fairness on DiscrimEval. This suggests that taking an expectation over confounder values may slightly reduce reward model accuracy, but it improves the overall fairness and robustness of the final policy. In contrast, the adversarial reward model performs worst, showing lower reward accuracy and weaker bias reduction, highlighting the advantage of our CPPO approach.

## USE OF LARGE LANGUAGE MODELS

We used a general-purpose LLM for copy-editing (e.g., grammar, phrasing, clarity) of draft text such as the abstract and introduction. The authors take full responsibility for all content in the paper, including any text that was edited with LLM assistance.

