# OpenReview forum: "Causal Proximal Policy Optimization"
_ICLR.cc/2026/Conference — ICLR 2026 Conference Withdrawn Submission_

### Official Review · Reviewer_8ceQ · 2025-10-15

**Soundness:** 2
**Presentation:** 1
**Contribution:** 2
**Rating:** 2
**Confidence:** 4

**Summary:**

This paper proposes Causal Proximal Policy Optimization (CPPO), a causal-inference–inspired framework designed to mitigate prompt-level confounding in Reinforcement Learning from Human Feedback (RLHF). The method introduces a Confounder Predictor that infers latent demographic attributes from prompts, a Confounder-Aware Reward Model that conditions rewards on these confounders, and a PPO objective that performs back-door adjustment by marginalizing over the predicted confounder distribution. The paper reports improved fairness on the DiscrimEval benchmark compared with Supervised Fine-Tuning, Vanilla PPO, and adversarial causal-reward baselines.

**Strengths:**

The paper addresses a timely and socially relevant problem—bias mitigation in Reinforcement Learning from Human Feedback (RLHF)—through the application of causal inference principles. The proposed pipeline is conceptually clean and integrates smoothly with widely adopted large language model (LLM) training frameworks such as PPO-based alignment.

**Weaknesses:**

The paper suffers from several notable weaknesses. First, the theoretical foundation lacks rigor — the causal identifiability assumptions (consistency, ignorability, and positivity) are directly borrowed from classical causal inference but are not justified in the context of high-dimensional prompt–response interactions in large language models. The predicted confounder, derived from prompts, does not satisfy the observability condition required for valid back-door adjustment, undermining the theoretical soundness of the approach. Second, the construction of demographic confounders using keyword-based filtering from the HH-RLHF dataset is problematic and likely introduces spurious correlations rather than genuine confounding effects, making the causal interpretation unreliable. Third, the experimental evaluation is limited, reporting only one metric (MaxDiscScore) without confidence intervals, ablations, or statistical significance testing, and lacking analysis of trade-offs between fairness and alignment performance. Moreover, the paper’s novelty is limited since the proposed method largely combines existing ideas from causal RLHF and causal reward modeling, contributing more as an implementation variant than a conceptual innovation. Finally, several implementation details critical to reproducibility, such as the interaction between the confounder predictor and the reward model, are deferred to the appendix, weakening the transparency and interpretability of the results.

**Questions:**

1. How does CPPO perform on standard RLHF alignment metrics (e.g., helpfulness, harmlessness) beyond bias reduction?
2. Would the method still hold if confounders are latent and unobservable, i.e., when no demographic annotations are available?
3. Please provide ablation results showing the effect of each component (predictor, back-door adjustment, marginalization).

---

### Official Review · Reviewer_YL5r · 2025-10-31

**Soundness:** 2
**Presentation:** 2
**Contribution:** 2
**Rating:** 2
**Confidence:** 3

**Summary:**

This paper proposes **Causal Proximal Policy Optimization (CPPO)**, a novel framework designed to mitigate bias in Reinforcement Learning from Human Feedback (RLHF). The authors argue that existing approaches—such as causal prompting or adversarial reward modeling—typically focus only on either prompt engineering or reward modeling, while overlooking **prompt-level confounders** (e.g., gender, age, race) that simultaneously influence both the model’s generated responses and the human-provided reward signals. CPPO addresses this through an end-to-end causal debiasing pipeline comprising three key steps:

1. **Confounder Predictor**: Predicts latent confounding variables (e.g., demographic attributes) directly from the prompt;
2. **Confounder-Aware Reward Model**: Trains the reward model conditioned on the predicted confounders;
3. **Backdoor-Adjusted Policy Optimization**: Marginalizes over the confounders within the PPO objective to obtain an unbiased interventional reward estimate.

Experiments on the DiscrimEval benchmark demonstrate that CPPO significantly outperforms baseline methods—including Supervised Fine-Tuning (SFT), standard PPO, and adversarial causal reward modeling—across three bias dimensions: race, gender, and age.

**Strengths:**

1. **Precise Problem Formulation**: The paper clearly identifies a critical gap in existing RLHF methods: the neglect of confounding bias along the causal pathway “prompt → confounder → response & reward.” This work fills an important void in applying causal debiasing to the full alignment pipeline.
2. **Simple yet Effective Approach**: CPPO achieves debiasing without requiring intermediate variables (e.g., Chain-of-Thought reasoning) or adversarial training; instead, it leverages backdoor adjustment alone, offering an elegant and efficient solution.

**Weaknesses:**

1. **Reliance on Confounder Labels**: Training requires human-annotated confounder labels (e.g., gender, race), which are typically unavailable in real-world RLHF datasets. Although the authors use keyword-based heuristics to approximate these labels, this may introduce noise or incomplete coverage.
2. **Confounder Prediction at Test Time**: During deployment, inaccurate prediction of confounders (e.g., misidentifying attributes in implicit prompts) can degrade the effectiveness of backdoor adjustment—a limitation explicitly acknowledged by the authors.
3. **High Computational Overhead**: Backdoor adjustment necessitates a separate forward pass of the reward model for each confounder category. When the number of categories is large (e.g., fine-grained race or religious groups), inference cost increases substantially.
4. **Single-Confounder Assumption**: The current framework handles only one confounder at a time (e.g., gender *or* age), and does not address multi-dimensional confounding (e.g., intersectional bias involving gender × race).
5. **Limited Generalization Validation**: Experiments are confined to DiscrimEval’s simple binary (yes/no) decision tasks. The method’s debiasing efficacy in more complex settings—such as open-ended generation or multi-step reasoning—remains unverified.

**Questions:**

1. How can CPPO be extended to simultaneously adjust for multiple confounders (e.g., gender + age + region)? Would this lead to a curse of dimensionality or fail to capture interaction effects among confounders?
2. After training with CPPO, does the model suffer any degradation in general capabilities? If so, how severe is this performance drop?

---

### Official Review · Reviewer_rEaH · 2025-10-31

**Soundness:** 2
**Presentation:** 3
**Contribution:** 2
**Rating:** 4
**Confidence:** 3

**Summary:**

This paper investigates the problem of social bias in large language models trained via RLHF, focusing on how demographic confounders in prompts can lead to biased model outputs. The authors propose Causal Proximal Policy Optimization (CPPO), which mitigates bias by explicitly modeling confounding variables and applying causal (back-door) adjustment during both reward modeling and policy optimization. They validate their method by filtering human preference data with demographic keywords and evaluating on the DiscrimEval benchmark, demonstrating that CPPO effectively reduces demographic bias compared to standard RLHF baselines.

**Strengths:**

1. The contribution of this paper is simple and clear.
2. It introduces causal inference techniques into the RLHF pipeline to address potential social biases present in reward models.
3. The effectiveness of the proposed approach is demonstrated through experiments.

**Weaknesses:**

1. I found Figure 1 somewhat confusing. According to the diagram, the prompt $s$ influences both the model output $a$ and the reward signal $r$ through the confounder $c$. While I understand that the intention is to illustrate that $c$ may simultaneously affect both $a$ and $r$, I am uncertain whether there should also be a direct connection from $s$ to $a$, to reflect the direct influence of the prompt content on model output. It would be helpful to clarify whether all effects of $s$ on $a$ are mediated exclusively through $c$, or if a direct effect is also considered.

2. Some symbols are used without prior definition. For example, in the “Data setting” section, $\mathcal{D}=\{(s_i,a_i^*,a_i^-,l_i,c_i)\}_{i=1}^N$ is introduced, but the meaning of $l_i$ does not appear to be defined.

3. In the experimental section, the model is validated on only one dataset, and the confounders considered are limited to race, gender, and age. This seems insufficient to fully demonstrate the robustness of the proposed approach. It would be beneficial to evaluate the model on additional datasets and with a broader range of confounders.

**Questions:**

Please check the Weaknesses.

---

### Note · Authors · 2025-11-18

**Comment:**

After considering the reviewers’ comments, we have chosen to withdraw the paper and thank the reviewers for their helpful insights.

**Withdrawal Confirmation:**

I have read and agree with the venue's withdrawal policy on behalf of myself and my co-authors.